# The First Complete Mitochondrial Genome of the Genus *Pachycondyla* (Formicidae, Ponerinae) and Insights into the Phylogeny of Ants

**DOI:** 10.3390/genes14081528

**Published:** 2023-07-26

**Authors:** Xingyu Lin, Nan Song

**Affiliations:** Henan International Laboratory for Green Pest Control, Henan Engineering Laboratory of Pest Biological Control, College of Plant Protection, Henan Agricultural University, Zhengzhou 450002, China; xingyulin666666@163.com

**Keywords:** poneroid ant, *Pachycondyla annamita*, mitochondrial genome, gene rearrangement, phylogeny

## Abstract

Ants are the standout group among eusocial insects in terms of their exceptional species richness and ecological dominance. The phylogenetic relationships among the group remain elusive. Mitochondrial genome sequences, as a kind of molecular marker, have been widely utilized in the phylogenetic analysis of insects. However, the number of ant mitogenomes published is still very limited. In this study, we utilized next-generation sequencing to determine the complete mitogenome of *Pachycondyla annamita* (Formicidae, Ponerinae). This is the first mitogenome from the genus *Pachycondyla*. Two gene rearrangements were identified in the mitogenome, the transposition of *trnQ* and *trnM* and the transposition of *trnV* and *rrnS*. The secondary structures of tRNAs were predicted. The tRNA genes *trnR* and *trnS1* lacked the dihydrouridine (DHU) arm, and the *trnE* lacked the TΨC (T) arm. Phylogenetic analyses of the mitochondrial protein-coding genes under maximum likelihood (ML) and Bayesian inference (BI) criteria resulted in conflicting hypotheses. BI analysis using amino acid data with the site-heterogeneous mixture model produced a tree topology congruent with previous studies. The Formicidae was subdivided into two main clades, namely the “poneroid” clade and the “formicoid” clade. A sister group relationship between Myrmicinae and Formicinae was recovered within the “formicoid” clade.

## 1. Introduction

Insect mitochondrial genome (mitogenome) sequences have been extensively utilized for the study of molecular evolution, population genetics, and phylogeny [1,2,3]. Mitogenomes, particularly in insects, exhibit unique characteristics such as high copy numbers per cell, high rates of evolution, limited or no recombination, and predominantly maternal inheritance [4,5,6]. Typically, the insect mitochondrial genome is a circular molecule encoding 37 genes, 13 protein-coding genes (PCGs), two ribosomal RNA (rRNA) genes, and 22 transfer RNA (tRNA) genes [1]. The latter two types of genes are involved in the translation of the PCGs [1]. In addition, there is a noncoding control region (also known as the A+T-rich region) containing elements relevant to the origin of replication and transcription of the mitogenome. The gene order in the insect mitogenome is also highly conserved. However, gene rearrangements are more frequently identified in some specific lineages, such as Hymenoptera.

Ants are the standout group among eusocial insects in terms of their exceptional species richness and ecological dominance. [7,8]. Ants are not only one of the most common insects found worldwide but are also well known to people. Almost all terrestrial ecosystems have been colonized by ants, except for tundra and cold ever-wet forests. Currently, over 14,000 species of ants have been described in the world [9], which are classified into over 300 genera and 9–21 subfamilies [10,11,12,13,14,15]. Although much progress has been made in illuminating the evolutionary history of ants, the phylogenetic relationships among subfamilies remain an open question [12,16].

Based on morphological characteristics, Brown (1954) identified nine subfamilies within the family Formicidae and classified them into two major lineages: the “myrmecioid complex” (consisting of Myrmeciinae, Pseudomyrmecinae, Dolichoderinae, and Formicinae) and the “poneroid complex” (composed of Cerapachyinae, Ponerinae, Myrmicinae, Dorylinae, and Leptanillinae). Taylor (1978) revised Brown’s (1954) classification scheme by transferring Myrmeciinae and Pseudomyrmecinae to the “poneroid complex”. The remaining lineages formed the “formicoid complex” [14,17]. Bolton (2003) classified Formicidae into 21 subfamilies based on morphology. A subsequent study by Brady et al. (2006) recognized 20 extant subfamilies for ants [10,18].

A study by Saux et al. (2004) using nuclear 28S rDNA sequences, divided the ant subfamilies into two main clades, one comprising Amblyoponinae, Ponerinae, and Proceratiinae and the other including Cerapachyinae, Dolichoderinae, Ectatomminae, Formicinae, Myrmeciinae, and Myrmicinae [19]. Ward et al. (2005) conducted a study using approximately 5.8 kilobases of sequence data from seven nuclear genes. Their findings supported the monophyly of the 20 existing subfamilies of ants proposed by Bolton (2003) [10,20]. Only Cerapachyinae was not recovered. However, relationships among the early diverging subfamilies of ants were ambiguous. Moreau et al. (2006), based on ~4.5 kb of sequence data from six gene regions, recovered 17 subfamilies as monophyletic, but the subfamilies Cerapachyinae and Amblyoponinae were not supported [15]. Brady et al. (2014), based on molecular data from 11 nuclear genes, merged members formerly assigned to the dorylomorph clade into a single subfamily, Dorylinae [13].

The formicoid clade, which includes the majority of ant species, has received support from multiple molecular studies [15,18,19,20,21]. This clade comprises the subfamilies Aneuretinae, Dolichoderinae, Dorylinae, Ectatomminae, Formicinae, Heteroponerinae, Myrmeciinae, Myrmicinae, and Pseudomyrmecinae [12]. A consensus on the relationships among subfamilies in the formicoid clade has been achieved: (Dorylinae, (((Myrmeciinae, Pseudomyrmecinae), (Aneuretinae, Dolichoderinae)), (Formicinae, (Myrmicinae, (Ectatomminae, Heteroponerinae)))). Vitellogenin gene duplication has been considered a synapomorphy of the formicoid clade [22,23]. Although the “formicoid” clade has gained molecular support, no specific morphological characteristic has been identified as a synapomorphy for this group.

A molecular study by Borowiec et al. (2019), based on 11 nuclear gene fragments for a total of approximately 7.5 kilobases of sequence data, reconstructed the internal phylogeny of Formicidae, with a special discussion on the rooting of the ant tree. Their results supported both *Martialis* (Martialinae) and Leptanillinae as being sister to all other ants [24]. Branstetter et al. (2017) used ultraconserved element (UCE) data to investigate the relationships among 16 ant subfamilies. Their results supported both the poneroid clade and the formicoid clade [25]. According to previous molecular studies, Leptanillinae was sister to the remaining ants. Recently, Romiguier et al. (2022) newly sequenced 65 ant genomes and reconstructed the phylogenetic relationships among the 17 ant subfamilies [9]. Their results provided strong support to the clade (*Martialis* + Leptanillinae) as the sister group to all other extant ants. In addition, the “poneroid” clade and the “formicoid” clade were also supported by the genome-wide data.

Compared with whole nuclear genome scale data, mitochondrial genome sequences are easier to assemble from next-generation sequencing data and to utilize in further phylogenetic analysis. Despite the enormous species richness and abundance of ants, only 83 complete or partial mitogenome sequences (as of 20 March 2023) for Formicidae have been published in GenBank. Sequencing additional mitogenomes will contribute to resolving the phylogeny of ants. In this study, we reconstructed the complete mitogenome of *Pachycondyla annamita* (Formicidae, Ponerinae), representing the first for this genus. Together with 83 existing ant mitogenome sequences, we conducted preliminary phylogenetic analyses of ant subfamilies under various data coding schemes and inference methods in an attempt to provide insights into the phylogeny of the group.

## 2. Materials and Methods

### 2.1. Taxon Sampling

The specimens of *P. annamita* were collected from Yao Mountain, Henan Province, China, at the geospatial coordinates N 33°47′21″, E 112°19′40″. They were immediately preserved in absolute ethanol and stored at −20 °C. Voucher specimens were deposited in the Entomological Museum of Henan Agricultural University. In addition to relying on external morphology for species identification, molecular sequence data were generated to confirm the identity of the specimens. Genomic DNA was extracted from the thoracic muscles using the TIANamp Genomic DNA Kit (Tiangen Biotech Co., Ltd., Beijing, China) following the manufacturer’s protocols. The quality and concentration of the total DNA were assessed using a NanoDrop 2000 spectrophotometer and 1% agarose gel electrophoresis, respectively.

### 2.2. Data Collection

The library construction and sequencing of the isolated DNA were completed at BGI Genomics Co., Ltd. (Wuhan, China) using the MGI2000 platform, with a strategy of 150 bp paired-end reads. A total of 2 Gb raw reads were generated. To ensure data quality, we utilized the NGS QC Toolkit v2.3.3 for quality control [26] under the default settings. The high-quality reads with an average Q20 > 90% and average Q30 > 80% were used for mitogenome assembly. For the assembly of mitogenome, we used the assembler GetOrganelle v1.7.5.2 [27]. The animal_mt database (-F animal_mt) was used to identify, filter, and assemble target-associated reads. In addition, we used Geneious R11 [28] to assemble the mitogenome, with the *cox1* gene sequence as reference. The parameters set for this task consisted of a minimum overlap identity set at 95%, ensuring that sequences have a minimum overlap of 50 bp. Furthermore, the algorithm allowed a maximum of 5% gaps per reading, with a maximum gap size limited to 20 bp. The iteration was performed up to 100 times, optimizing the sensitivity according to custom specifications. The mapping to the mitochondrial contig was performed using Geneious R11 [28].

We used the MITOS webserver to annotate the newly sequenced mitogenome of *P. annamita* [29]. The parameter settings were as follows: Genetic Code—05-inverterbrate; Reference—RefSeq 89 Metazoa. All other parameters were set to default values. We employed two programs, MITOS [29] and ARWEN [30], to infer the tRNA genes. The secondary structures of tRNA genes were recreated using Adobe Illustrator CS. The circular mitogenome map of *P. annamita* (Figure 1) was generated in the Mtviz online webserver (http://pacosy.informatik.uni-leipzig.de/mtviz (accessed on 16 March 2023)). The line maps of gene order of ant mitogenomes were drawn by PhyloSuite [31]. The obtained mitochondrial genome sequence has been successfully submitted to GenBank and assigned the accession number OQ629337.

We used MEGA X to calculate the nucleotide composition of the mitogenome sequences [32]. AT- and GC-skew values were calculated following the formula from [33]. The tool CREx (http://pacosy.informatik.uni-leipzig.de/crex/form (accessed on 16 March 2023)) [34] was employed to identify the most parsimonious explanation for the gene arrangement observed in *P. annamita*. The common intervals parameter was utilized for measuring the distances and comparing the gene orders. 

### 2.3. Sequence Alignment

To align the protein-coding genes, we employed codon-based multiple alignments using the MAFFT algorithm [35] with the invertebrate mitochondrial genetic code. Prior to back translating to nucleotides, we utilized trimAl v1.4 to remove any poorly aligned sites from the protein alignment [36]. Using the “E-INS-i” iterative refinement method, we performed separate alignments of rRNA genes and tRNA genes on the MAFFT server [35]. The resulting alignments were concatenated with FASconCAT-G_v1.04 [37] to compile the following datasets: (1) PCG_nt, nucleotide alignment including 13 protein-coding genes and (2) PCG_aa, amino acid alignment including 13 protein-coding genes. Substitution saturation of sequences was assessed with DAMBE 6 [38,39].

### 2.4. Phylogenetic Analysis

As ingroups, we included 84 species representing seven subfamilies of Formicidae (Appendix A). In addition, we selected two species from Vespidae as outgroups. Phylogenetic analyses were perform using maximum likelihood (ML) and Bayesian inference (BI) approaches. ML analyses were conducted using IQ-TREE 2.2.2 [40]. Nucleotide sequences were partitioned by codons and genes, whereas amino acid sequences were partitioned by genes. ModelFinder [41] was used to select the best-fitting substitution models for each partition. Branch support (BS) values were calculated using ultrafast bootstrap [42], with 10,000 replicates.

BI analyses were performed using PhyloBayes MPI (v1.8) [43]. For nucleotide sequences, we employed the CAT-GTR model. The CAT-mtART model was used for phylogenetic analyses of amino acid sequences. To thoroughly explore the parameter space, we ran two parallel chains with a combined total length of 10,000 cycles. We evaluated convergence of the chains by examining the difference in frequency for all of their bipartitions (maxdiff < 0.3). A majority-rule consensus tree was computed using the program *bpcomp* implemented in PhyloBayes [43], with the first 20% of trees discarded. Branch support was assessed using the posterior probability (PP) values.

### 2.5. Hypothesis Testing

In order to assess different signals for the relationships among the formicoid clade including Pseudomyrmecinae, Proceratiinae, Amblyoponinae, Ponerinae, Dolichoderinae, Myrmicinae, and Formicinae, we performed a four-cluster likelihood mapping (FcLM) analysis [44,45]. This analysis aimed to explore alternative patterns and evaluate the robustness of these relationships. The dataset used for the analysis consisted of 86 taxa and was specifically focused on nucleotide sequence data and amino acid sequence data. IQ-TREE [40] with the mtART model was employed as the computational tool to conduct the analysis. By employing this approach, we aimed to gain insights into the evolutionary history and genetic relationships among these taxa.

## 3. Results

### 3.1. Genome Sequencing and Assembly

Approximately 2 Gb of raw PE data were produced for *P. annamita*. Of these, 16,947,812 clean reads were obtained after quality trimming, and these were used in the subsequent mitogenome assembly. Two assemblers yielded an identical scaffold for the *P. annamita* mitogenome. The length of mitogenome was 15,466 bp. The total number of reads mapped across the assembled mitogenome was 85,990, accounting for 0.5% of the genomic read pool obtained. The mean base coverage of the mitogenome was 736-fold. Read mapping showed a basically uniform distribution of reads along the mitogenome.

### 3.2. Characteristics of the P. annamita Mitogenome

The complete mitogenome of *P. annamita* contained the typical set of 37 genes and a control region (Figure 1). Compared with the putative ancestral insect mitogenome, gene rearrangements were found in two regions in the *P. annamita* mitogenome: (1) *trnQ* and *trnM* transposition and (2) *trnV* and *rrnS* transposition (Figure 2). 

The nucleotide compositions of the entire mitogenome were as follows: A = 41.1%; T = 44.4%; C = 8.2%; and G = 6.3%, with an overall A+T content of 85.5%. The mitogenome was significantly biased towards A+T. On the major strand, the AT-skew and GC-skew were −0.038 and −0.136, respectively. On the minor strand, both AT-skew and GC-skew were positive (0.038 and 0.136). Nine PCGs (*atp6*, *atp8*, *cox1*, *cox2*, *cox3*, *cytb*, *nad2*, *nad3*, and *nad6*) were coded on the major strand. The remaining four PCGs (*nad1*, *nad4*, *nad4L*, and *nad5*) were coded on the minor strand. The PCGs on the major strand had an A+T content of 82.8%, whereas those on the minor strand had an A+T content of 85.9%. The average A+T contents of rRNAs and tRNAs were 85.5% and 87.1%, respectively. The putative control region had the highest A+T content (94.5%). 

All 13 PCGs used the standard ATN (ATG, ATT, and ATA) initiation codons and terminated with the stop codons of TAA. Leu2, Ile, Phe, Met, and Asn were the most frequently used amino acids. The RSCU (relative synonymous codon usage) is shown in Figure 3. It is obvious that the codons ending with A or T (green and blue columns) outnumber those ending with G or C (purple and red columns). The lengths of tRNAs ranged from 59 (*trnR*) to 78 bp (*trnT*). The secondary structures of tRNAs are presented in Figure 4. Nineteen of the 22 tRNAs can be folded into typical cloverleaf secondary structures. The tRNA genes *trnR* and *trnS1* lacked the dihydorouridine (DHU) arm due to unmatched base pairs, and the TΨC (T) arm was missing in the *trnE*.

### 3.3. Phylogenetic Inference

We analyzed concatenated alignments of PCGs using probabilistic methods of tree reconstruction with (i) a site-homogeneous model of sequence evolution in the ML (GTR model) framework and (ii) a site-heterogeneous mixture model in a Bayesian framework (CAT model). Figure 5 and Figure 6 depict the phylogenetic relationships inferred from these analyses. Across analyses, the subfamilies with multiple exemplars included in this study were consistently recovered: Ponerinae, Pseudomyrmecinae, Dolichoderinae, Myrmicinae, and Formicinae. A sister group relationship between Myrmicinae and Formicinae was strongly supported (BS > 90, PP > 0.9). The newly sequenced *Pachycondyla annamita* was placed within Ponerinae and was sister to *Brachyponera chinensis* (BS = 100, PP = 1 or 0.96). 

The branching order within Formicidae was different between ML and BI analyses. The main conflict lay in the placement of Pseudomyrmecinae. ML analyses from both nucleotide sequence data and amino acid sequence data yielded an identical tree topological structure for the relationships among subfamilies. Two main clades were retrieved in the ML trees. One included the Pseudomyrmecinae alone, whereas the other comprised the remaining subfamilies. Within the latter clade, the subfamilies Proceratiinae, Amblyoponinae, and Ponerinae clustered together (PCG_nt-ML: BS = 61, PCG_aa-ML: BS = 99). This group corresponded to the poneroid clade and diverged firstly in the large clade. The Dolichoderinae split secondly and the following were the sister groups of Myrmicinae and Formicinae. In the BI tree from PCG_aa (Figure 6), the Formicdae could also be subdivided into two major clades. However, the subfamilies Proceratiinae, Amblyoponinae, and Ponerinae grouped in a clade, which was sister to the rest of the subfamilies. In this BI tree, the formicoid clade was recovered. The Pseudomyrmecinae was nested within the formicoid clade and placed between the Dolichoderinae and the clade (Myrmicinae + Formicinae). The BI analysis from the dataset PCG_nt also recovered the monophyly of the formicoid clade. However, the Pseudomyrmecinae was sister to all the other three formicoid subfamilies. In addition, the relationships among the three subfamilies of the poneroid clade were ambiguous in the BI tree of PCG_nt. FcLM analyses showed more phylogenetic signal for the formicoid clade including Pseudomyrmecinae, Dolichoderinae, Myrmicinae, and Formicinae in both nucleotide sequence data and amino acid sequence data (Figure 7). 

## 4. Discussion

### 4.1. Gene Rearrangement

Comparisons of gene order revealed that almost all existing ant mitogenomes have gene rearrangements. The gene rearrangements mainly occur adjacent to the control region. Specifically, the position of *trnM* changed in all ant mitogenomes. Mapping the gene order of ant mitogenomes to the phylogenetic tree inferred from the present mitogenome sequences, we found that most members of Myrmicinae shared a gene arrangement of *rrnS–trnV*. This pattern can serve as a potential synapomorphy of all Myrmicinae. Compared with other species of Ponerinae, *P. annamita* displayed two novel gene rearrangement patterns: *trnM–trnQ* and *rrnS–trnV*. Sequencing more ant mitogenomes is expected to further test these results. 

### 4.2. Phylogeny of Ants

Applying ML and Bayesian inference methods under different sequence evolution models to our data resulted in conflicting views on the phylogenetic relationships of ants. Incongruence between the trees yielded from ML and Bayesian inferences mainly concerned the placement of Pseudomyrmecinae. ML analyses placed Pseudomyrmecinae as an independent clade and sister to all other ants, whereas BI analyses clustered Pseudomyrmecinae with Dolichoderina, Myrmicinae, and Formicinae. Comparing the two ML trees, the nucleotide and amino acid sequences produced an identical tree topology. However, the tree obtained based on the nucleotide sequence dataset displayed generally lower bootstrap values than the one obtained based on the amino acid sequence dataset, especially at the deeper nodes. This was likely due to the substitution saturation in the nucleotide sequence data (Appendix A). In particular, the third codon positions (*Iss* > *Iss.cAsym*) tended to weaken the phylogenetic signal at deeper nodes, resulting in a reduction in bootstrap support. Some authors have questioned the robustness of tree reconstructions based on mitochondrial datasets as a potential artifact of the analysis, such as sparse taxon sampling, type of data, inference methods, and peculiarities of the evolution of mitogenome [29,46,47,48]. To improve the phylogenetic analysis, we also used the site-heterogeneous mixture model, which has been shown to reduce the effects of compositional heterogeneity and mutational bias and fit the evolution of insect mitogenome sequences better than site-homogeneous models [49,50,51,52,53]. Therefore, we considered the tree from Bayesian inference of amino acid sequence data as our best estimate of the phylogenetic relationships of ants and focus the following discussion on this tree topology.

The non-formicoid clades included the poneroid ants and the subfamily Leptanillinae. Regarding the placement of Leptanillinae, several previous molecular studies have tended to agree that it is an early-diverging clade in Formicidae [9,24,25,54,55]. The “poneroid” clade comprised five subfamilies: Agroecomyrmecinae, Amblyoponinae, Paraponerinae, Ponerinae, and Proceratiinae [15,56,57]. Relationships among the poneroid subfamilies remained elusive. Brady et al. (2006) recovered Proceratiinae as sister to all other poneroid lineages [18]. Amblyoponinae was placed between Proceratiinae and the remaining poneroids. Moreau et al. (2006) clustered Amblyoponinae and Proceratiinae in a clade [15]. Branstetter et al. (2017) recovered Proceratiinae and Ponerinae as sister groups [25]. Amblyoponinae and Apomyrminae grouped together, but this relationship received no statistical support. In the analyses of Borowiec et al. (2019) [24], the placements of Proceratiinae and (Amblyoponinae+Apomyrminae) varied across analyses. Romiguier et al. (2022) placed Ponerinae as sister to all other poneroid lineages and also retrieved Amblyoponinae and Apomyrminae in a clade [9]. Due to the unavailability of mitogenome sequences for Agroecomyrmecinae and Paraponerinae, our analyses included only the subfamilies Amblyoponinae, Ponerinae, and Proceratiinae. The three subfamilies formed a monophyletic group regardless of the inference methods and data coding strategies. Moreover, Amblyoponinae and Proceratiinae formed a sister group relationship. Both of them were sister to Ponerinae. These two sister group relationships were robustly supported by amino acid data under the ML and Bayesian inference frameworks.

One of the most important groups recovered by the present analyses of mitogenome sequences is the group corresponding to the “formicoid” clade. Although the monophyly of the “formicoid” clade was not recovered by the previous morphological analyses, molecular studies have provided strong support for the group [9,15,18,19,21,24,25,58]. Brown (1954) subdivided ants into two major lineages, one was the “poneroid complex” and the other was the “myrmecioid complex” [14]. The latter complex comprised the subfamilies Myrmeciinae, Pseudomyrmecinae, Dolichoderinae, and Formicinae. This complex was confirmed by the present mitogenome sequence data. Within this complex, Myrmicinae and Formicinae formed a sister group relationship. This arrangement was consistent with previous studies [15,18]. More recent molecular studies recovered an alternative hypothesis on the relationships of Myrmicinae and Formicinae: a clade of (Heteroponerinae + Ectatomminae) was recovered as the sister group of Myrmicinae; this large clade was sister to Formicinae [9,24]. In future studies, additional sampling of Heteroponerinae and Ectatomminae are necessary to test the relationships among the formicoid subfamilies.

## 5. Conclusions

Using next-generation sequencing, we determined the complete mitogenome of the poneroid ant *P. annamita*. This is the first representative from the genus *Pachycondyla*. The *P. annamita* mitogenome comprised the set of 37 mitochondrial genes. Two gene rearrangements were found in this mitogenome: the transposition of *trnQ* and *trnM* and the transposition of *trnV* and *rrnS*. The gene arrangement was distinguished from the existing poneroid ant mitogenomes. Phylogenetic analysis from amino acid data under Bayesian inference using the site-heterogeneous mixture model yielded a relationship for ants congruent with previous studies. The Formicidae was subdivided into two main lineages that corresponded to the “poneroid” clade and the “formicoid” clade. Within the “formicoid” clade, a sister group relationship between Myrmicinae and Formicinae was recovered by the present mitogenome data. The study demonstrated that the site-heterogeneous mixture model can enhance the accuracy of phylogenetic reconstruction by incorporating information from different sites within mitochondrial protein-coding genes. This approach has been shown to be particularly effective at resolving conflicts between known genetic relationships and conflicting data. By providing a more comprehensive representation of evolutionary history, this method has the potential to greatly advance our understanding of mitochondrial evolution and the diversity of life on earth.

## Figures and Tables

**Figure 1 genes-14-01528-f001:**
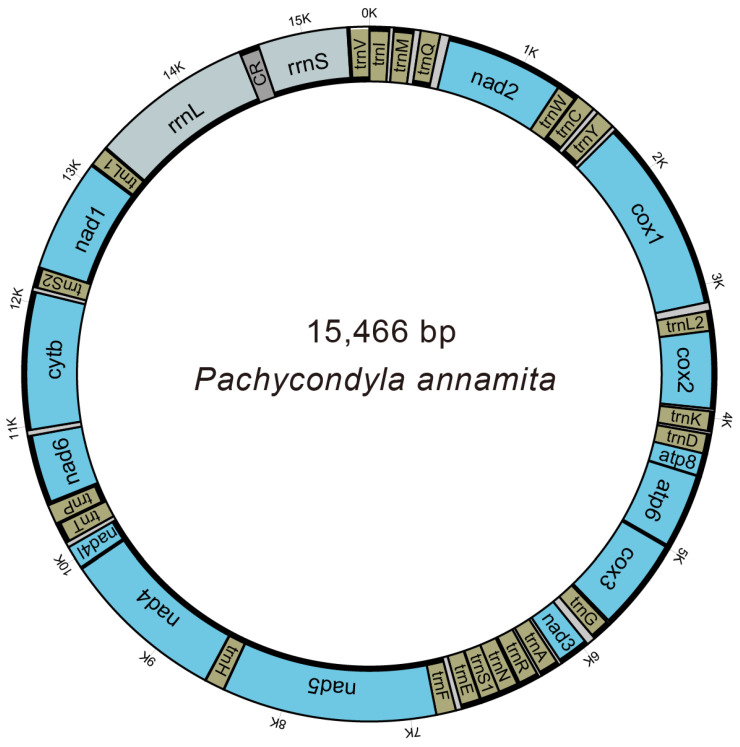
Organization of the mitogenome of *Pachycondyla annamita.* Bold lines indicated the coding strands.

**Figure 2 genes-14-01528-f002:**
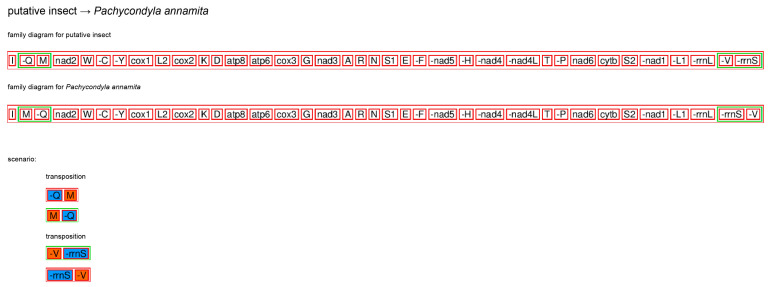
The gene rearrangement events inferred from the CREx analysis for the mitogenome of *Pachycondyla annamita*.

**Figure 3 genes-14-01528-f003:**
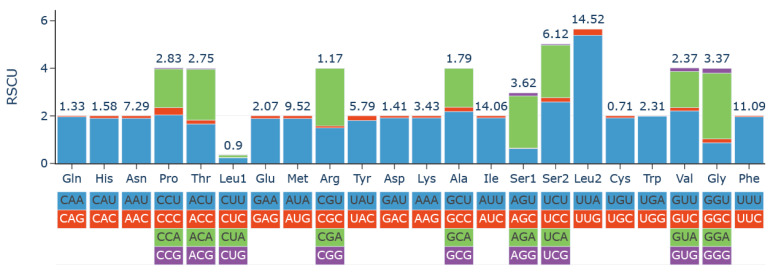
Codon usage of the 13 mitochondrial protein-coding genes of *Pachycondyla annamita*. RSCU: relative synonymous codon usage. Codon families are indicated on the X-axis and frequency of RSCU on the Y-axis.

**Figure 4 genes-14-01528-f004:**
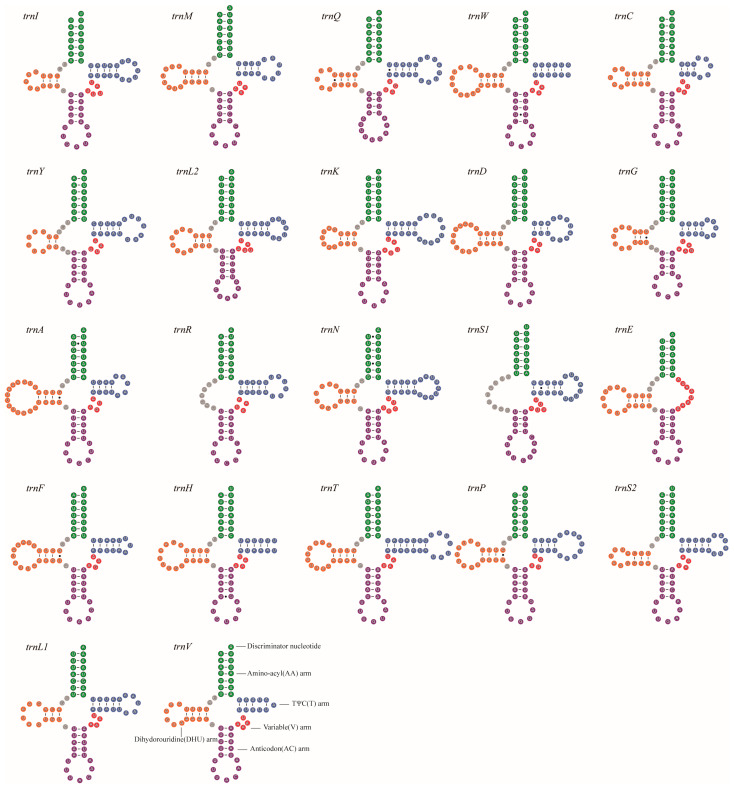
The secondary structures of tRNA genes inferred for the mitogenome of *Pachycondyla annamita.* Watson–Crick pairs are indicated by lines, and wobble GU pairs are indicated by dots. The non-canonical pairs are not marked.

**Figure 5 genes-14-01528-f005:**
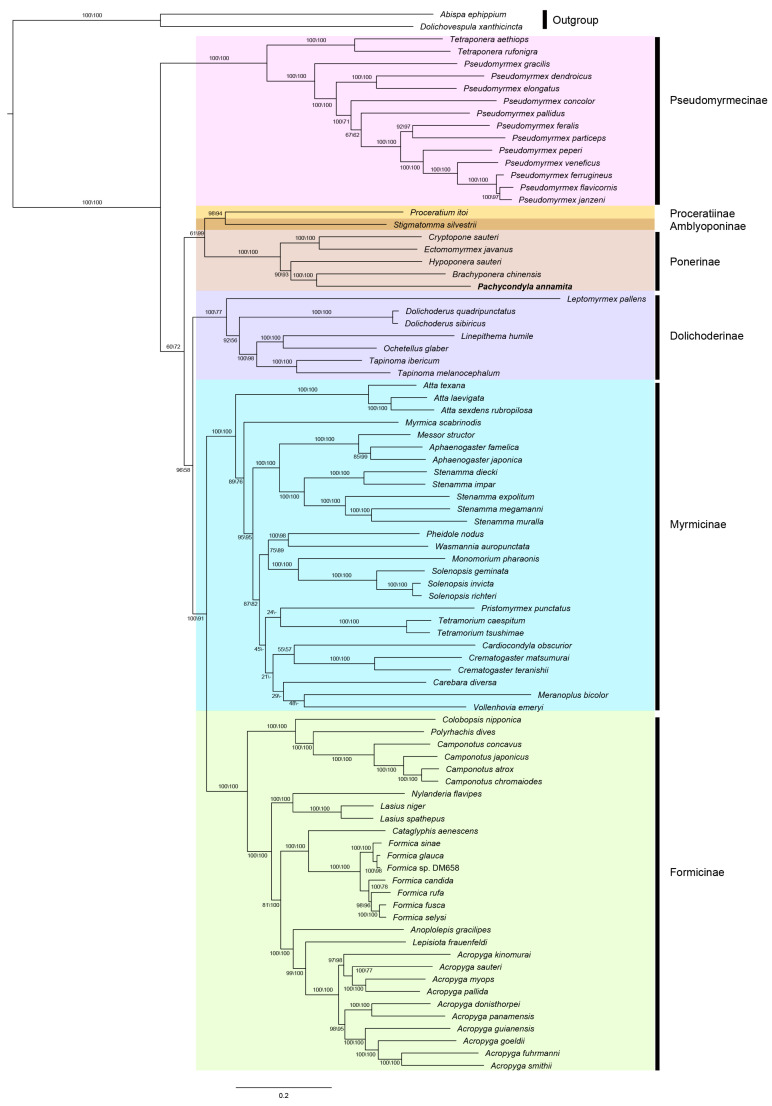
ML phylogenetic tree inferred in the IQ-TREE using the nucleotide sequences of 13 PCGs. Numbers at the nodes are ultrafast bootstrap values (left: PCG_nt, right: PCG_aa). Dash indicates the relationships not supported by the amino acid data. Scale bar represents substitutions/site.

**Figure 6 genes-14-01528-f006:**
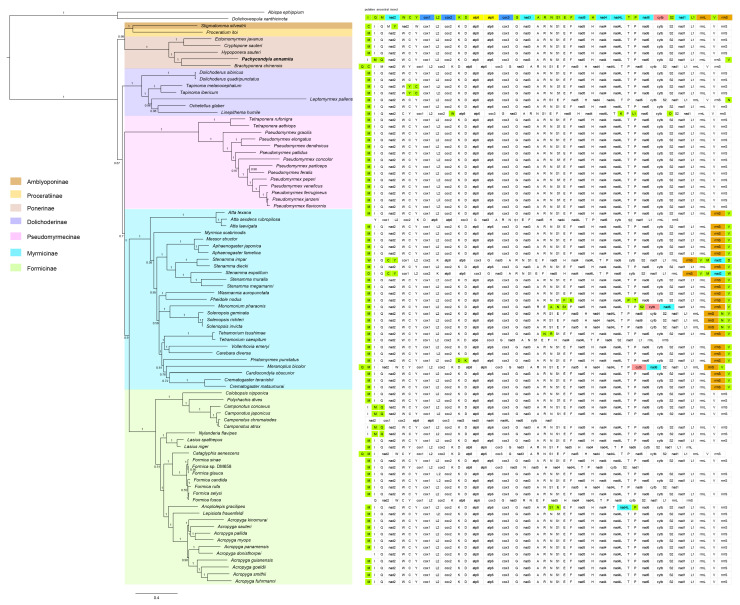
Bayesian phylogenetic tree inferred in PhyloBayes-MPI using the amino dataset PCG_aa (**right**) and the gene orders of the ants’ mitogenomes (**left**). Numbers at the nodes in the tree are Bayesian posterior probabilities. Scale bar represents substitutions/site.

**Figure 7 genes-14-01528-f007:**
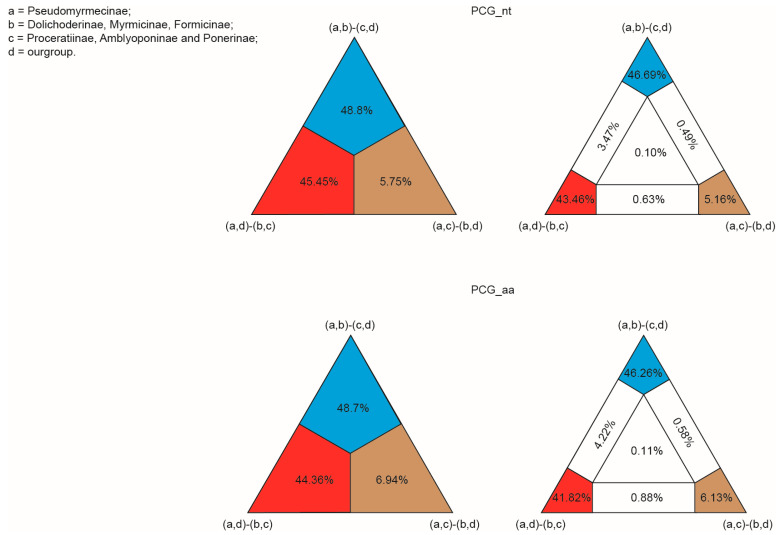
Results of the FcLM analyses on the datasets PCG_nt and PCG_aa for the alternative phylogenetic hypotheses.

## Data Availability

The newly obtained mitogenome sequence from this study has been deposited in GenBank under the accession number OQ629337.

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
