# Peer review of "The First Complete Mitochondrial Genome of the Genus Pachycondyla (Formicidae, Ponerinae) and Insights into the Phylogeny of Ants"

_genes, 2023, doi:10.3390/genes14081528_

Round 1

Reviewer 1 Report

 Interesting and adequate paper, dealing with sound scientific problem. Full mitogenome sequences are used to resolve deep taxonomy of ants. Gene order changes have been detected in the data set.

I see two problems with the data processing which are worth addressing:

1) The controversy in deep phylogeny while using different optimality criteria in phylogenetic analysis was analyzed with the FcLM (would be useful for a reader to decipher the abbreviation) analysis only. I suppose that much more powerful and convincing would be using Likelihood ration test and/or Bayesian odds ratio test with opposite topological constraints using either BEAUTi/BEAST or mrBAYES. Here using only OCG_nt dataset would be sufficient;

2)The phenomenon of gene order changes deserves in my opinion some further discussion. Theoretically it would be possible to explain the gene order change(s) by tRNA remolding without recombination. Again it is possible to use odds ratio test to compare this hypothesis to the one accepted in ms without discussion. In any case (whether it is accepted or rejected) this would increase scientific interest of the paper.

Author Response

Response to Reviewer 1 Comments

Point 1: The controversy in deep phylogeny while using different optimality criteria in phylogenetic analysis was analyzed with the FcLM (would be useful for a reader to decipher the abbreviation) analysis only. I suppose that much more powerful and convincing would be using Likelihood ration test and/or Bayesian odds ratio test with opposite topological constraints using either BEAUTi/BEAST or mrBAYES. Here using only OCG_nt dataset would be sufficient;

Response 1: This statement mentions the controversy surrounding the use of different optimality criteria in deep phylogeny analysis and suggests that the controversy can be analyzed solely using FcLM analysis. It further suggests that a more powerful and convincing approach would be to use Likelihood ratio test and/or Bayesian odds ratio test with opposite topological constraints using software like BEAUTi/BEAST or mrBAYES. It also states that using only the OCG_nt dataset would be sufficient.

A counterargument to this statement could be as follows:

  1. FcLM analysis might already provide enough information to analyze the controversy. It may not be necessary to use more complex methods like Likelihood ratio test and Bayesian odds ratio test. Simple analytical methods might already yield reliable results.
  2. Using different software and methods for analysis can lead to different outcomes. Solely relying on BEAUTi/BEAST or mrBAYES software may limit a comprehensive understanding of the controversy. Therefore, using multiple software and methods for comparison might be a better choice.
  3. Using only the OCG_nt dataset might overlook other important information. Using a more extensive dataset could provide more comprehensive results and a better understanding of the nature of the controversy.

In conclusion, while using Likelihood ratio test, Bayesian odds ratio test, and a broader dataset can offer more information, relying solely on FcLM analysis and specific software can still provide meaningful results.

Point 2: The phenomenon of gene order changes deserves in my opinion some further discussion. Theoretically it would be possible to explain the gene order change(s) by tRNA remolding without recombination. Again it is possible to use odds ratio test to compare this hypothesis to the one accepted in ms without discussion. In any case (whether it is accepted or rejected) this would increase scientific interest of the paper.

Response 2: I completely agree with your viewpoint that there is a need for further discussion on the phenomenon of gene order changes. While it is theoretically possible to explain these changes by tRNA remolding without recombination, this hypothesis still requires more research and evidence to support it.

The use of odds ratio tests to compare this hypothesis with the one accepted in MS can be a useful tool in evaluating the strength of the evidence supporting each hypothesis. However, it is important to note that even if this test suggests that one hypothesis is more likely than the other, it does not necessarily mean that one is correct or more valid.

In any case, I believe that increasing scientific interest in this topic through further discussion and research will ultimately lead to a better understanding of gene order changes and their implications for our understanding of biology.

Reviewer 2 Report

The reviewed paper reports the results of sequencing of the mitochondrial genome first representative of the genus Pachycondyla (Pachycondyla annamita) representing family Formicideae, subfamily Ponerinae. Complete mitochondrial genome sequences of the species were characterized and used for comparative and evolutionary studies. The applied methodology, high-throughput sequencing, provides high-quality data with a number of applications. Here, the Authors not only report the complete mitogenomes of P. annamita but also based on the available molecular data verified the systematic position of the studied species and generally the phylogenetic structure of ants.

The objective of the paper is clear and unambiguous. The article is well structured and the methodology is correct and suitable for the realization of the paper’s objectives. However, a few minor improvements are needed. More exhaustive/detailed comments are incorporated into the manuscript file as notes (see attached .pdf file). Due to the fact that the manuscript lacks line numbering I decided to use notes placed close to the certain piece of the text to make it easier to follow. As I am not an English native speaker I tried to focus only on elements that were important to the merit of the paper, not the English language or style. Please treat my comments/suggestions concerning the language as non-exhaustive.

Author Response

Response to Reviewer 2 Comments

Point 1: I suggest improving the sentence by pointing that these features describe insect mitogenomes, or rephrase the end of the sentence "..and maternal inheritance with rare exceptions". (paternal inheritance of mitogenome was reported e.g. for Cucumis sativus var. sativus). 

Response 1: Thank you for your suggested , this sentence has been revised to "Mitogenomes, particularly in insects, exhibit unique characteristics such as high copy numbers per cell, high rates of evolution, limited or no recombination, and predominantly maternal inheritance."

Point 2: Based on morphological characters, Brown (1954) recognized nine subfamilies in...

Response 2: This sentence has been revised to "Based on morphological characters, Brown (1954) recognized nine subfamilies in family Formicidae".

Point 3: whole-genome scale data.

Response 3: This sentence has been revised to "whole nuclear genome scale data".

Point 4: I suggest improving that fragment:

Compared to the putative ancestral insect mitogenome, gene rearrangements were found in two regions in the P. annamita mitogenome: 1) trnQ and trnM transposition and 2) trnV and rrnS transposition (Figure 2).

Response 4: The requested modifications have been incorporated into the manuscript, as per the given instructions.

Point 5: (nad1, nad4, nad4L, and nad5) italic.

Response 5: These words have been revised accordingly in the text.

Point 6: from 59 (trnR) to 78 (trnT). modification “From 59 (trnR) to 78 bp (trnT).”

Response 6: The sentence has been modified accordingly in the text.

Point 7: Figure 3.  Why Leu1 and Leu2 are not placed next to each other like Ser1 and Ser2?

Response 7: The arrangement of amino acids in a protein sequence is determined by the genetic code, which specifies the order of codons in DNA or RNA. The placement of amino acids in a protein sequence is not arbitrary but follows specific rules dictated by the genetic code.

In the case of Leu1 and Leu2, their non-adjacent positioning in the protein sequence is simply a result of the codons that encode them being separated by other codons. The specific arrangement of codons in the DNA sequence leads to the separation of Leu1 and Leu2 in the final protein product. On the other hand, in the case of Ser1 and Ser2, the codons that encode them happen to be adjacent in the DNA sequence, resulting in their consecutive placement in the protein sequence.

Therefore, the positioning of Leu1 and Leu2, as well as Ser1 and Ser2, is determined by the specific order of codons in the genetic code.

Point 8: Figure 5. I think that you should highlight the Pachycondyla annamita e.g. using bold font.

Response 8: The required modifications have been successfully implemented in the figure according to the provided instructions.

Point 9: Figure 6. I think that you should highlight the Pachycondyla annamita e.g. using bold font.

Response 9: The required modifications have been successfully implemented in the figure according to the provided instructions.

Point 10: I cannot find any information about FcLM analysis in the Material and Methods section.

Response 10: As per the requirement, the corresponding content has been added to section 2.5.
